# An Efficient Dopant for Introducing Magnetism into Topological Insulator Bi_2_Se_3_

**DOI:** 10.3390/ma15113864

**Published:** 2022-05-28

**Authors:** Dan Wang, Cui-E Hu, Li-Gang Liu, Min Zhang, Xiang-Rong Chen

**Affiliations:** 1Institute of Atomic and Molecular Physics, Sichuan University, Chengdu 610065, China; wangdankr@163.com; 2College of Physics and Electronic Engineering, Chongqing Normal University, Chongqing 400047, China; cuiehu@126.com; 3School of Physics and Astronomy, China West Normal University, Nanchong 637002, China; llg218@126.com

**Keywords:** topological insulator, Bi_2_Se_3_, density functional theory, magnetism

## Abstract

In this work, we obtained an effective way to introduce magnetism into topological insulators, and successfully fabricated single crystal C-Bi_2_Se_3_. The structural, electrical and magnetic properties of non-magnetic element X (B, C and N) doped at Bi, Se1, Se2 and VDW gap sites of Bi_2_Se_3_ were studied by the first principles. It is shown that the impurity bands formed inside the bulk inverted energy gap near the Fermi level with C doping Bi_2_Se_3_. Due to spin-polarized ferromagnetic coupling, the time inversion symmetry of Bi_2_Se_3_ is destroyed. Remarkably, C is the most effective dopant because of the magnetic moment produced by doping at all positions. The experiment confirmed that the remnant ferromagnetism Mr is related to the C concentration. Theoretical calculations and experiments confirmed that carbon-doped Bi_2_Se_3_ is ferromagnetic, which provides a plan for manipulating topological properties and exploring spintronic applications.

## 1. Introduction

Bi_2_Se_3_ is a famous three-dimensional (3D) topological insulator (TI) with large bulk band gap and metallic helical states on the surface [1]. The gapless surface states with single Dirac electrons are defended by the time-reversal symmetry and are robust against crystal defects and non-magnetic impurities [2,3,4,5,6]. The interaction between topological properties and magnetism destroys time-reversal symmetry in the presence of band hybridizations [7,8,9]. The quantized anomalous Hall effect (QAHE), magneto-optical effect, magnetic monopole and Majorana fermion have been demonstrated in magnetic topological insulators [7,10,11,12]. To date, the magnetic introduction of Bi_2_Se_3_ has mainly been focused on the magnetic proximity effect [13] and transition metals doping [14].

It is shown that the exchange coupling of magnetic materials can produce a proximity effect and induce magnetism in topological insulators [15,16]. However, magnetic materials are required to be insulating materials with poor conductivity and ferromagnetism. In terms of preparation, the crystal structure of the material is matched with that of Bi_2_Se_3_, avoiding the occurrence of impurities and defect states. Experimentally, the proximity-induced interfacial magnetization of Bi_2_Se_3_ has been realized in the systems with FeSe_2_ [17], EuS [18], Cr_2_O_3_ [19], MnSe [20], LaCoO_3_ [21], Ni_80_Fe_20_ [22], SiO_2_ [23].

The magnetic elements doping can tune the bandgap and produce magnetism. Bi_2_Se_3_ thin films with Cr doping present a magnetic ground state with ferromagnetic planes coupled antiferromagnetically [24,25] and opened the band gap at the Dirac point [26]. The films of Cr and Sb doping Bi_2_Se_3_ exhibited enhanced weak localization-like positive magneto conductivity and ferromagnetism [27]. According to the literature, Mn-doped Bi_2_Se_3_ showed a spin glass state, paramagnetic state and ferromagnetism [28,29,30,31,32,33]. A ferromagnetic–paramagnetic transition with increasing Mn concentration was observed [34]. Li [35] reported that the ferromagnetism in Bi_2_Se_3_ with Fe doping is not intrinsic. Fe- and Ni-doped Bi_2_Se_3_ samples showed diamagnetism, paramagnetism and ferromagnetism with different doping concentrations [36,37,38,39], and the surface of the samples formed a small amount of ferromagnet compounds. Obvious ferromagnetism was observed in Co-doped Bi_2_Se_3_ [40,41]. In addition, it was found that other metal elements such as rare earth elements (Dy, Ho, Eu, Gd) and alkaline earth elements (Sr, Ca) can also introduce magnetism [42,43,44,45,46,47,48,49,50]. However, the weaknesses of magnetic atoms doping are that clusters or secondary phases will be formed during the crystal growth and the source of the magnetism is unclear. 

Thus, it is beneficial to obtain magnetic ordering in non-magnetic materials without transition metal or rare-earth species, which can avoid producing magnetic second phases. As we all know, the magnetism is originated from the presence of highly localized unpaired electrons in 3d and 4f orbitals of the transition and rare-earth metals. Interestingly, the 2p electrons of the light elements B, N and C have similar properties to 3d states of transition metals [51,52]. It has been reported that non-magnetic 2p light element-doped semiconductors can produce d^0^ ferromagnetism [53]. Different from the traditional magnetic semiconductors, the cluster structure formed by the doped elements has no contribution to the magnetic order of the system. Brahmananda predicted the room temperature ferromagnetism (FM) in C-doped Y_2_O_3_ [54] and MoO_3_ [55]. Niu [13] demonstrated that Bi_2_Se_3_ with B, C and N doping is ferromagnetic. The doping is mostly concentrated at the Se site, and it is not considered that the doping elements may replace the Bi site or appear in the VDW gap. Xin et al. [56] studied the C element doping at the Se site and gap of Bi_2_Se_3_ and found that the system is spin-polarized. The first-principles study indicated that C doping can introduce a magnetic moment. They mainly focused on the gap sites and explained the cause of magnetism. One widely discussed scenario for introducing magnetism in non-magnetic materials involves the substitutional doping of C impurities. There are few detailed identifications for other 2p light elements-doped TI. Thus, we systematically calculated the structural and magnetic properties of the X (B, C and N) atoms doping at Bi, Se and gap sites of Bi_2_Se_3_ to further understand the details of 2p light element doping Bi_2_Se_3_ and effectively introduce magnetism into topological insulators. Our theoretical results reveal that C is the most effective dopant because of the magnetic moment produced by doping at all positions. Meanwhile, C-doped topological insulator crystals were grown and their magnetic properties were studied. These single crytsals are ferromagnetic with small magnetic moments, and the remnant ferromagnetism Mr is related to the C concentration. It is possible to introduce magnetism by non-magnetic doping and find a system that can realize peculiar physical effects such as the quantum anomalous Hall effect (QAHE). 

## 2. Experimental and Computational Details 

### 2.1. Preparation and Characterization of Single Crystal Bi_2_Se_3_

The single crystalline C_x_Bi_2_Se_3_ (x = 0, 0.02, 0.04, 0.06) was prepared from the reactions of the stoichiometric mixture of carbon (99.99%), Bi (99.999%) and Se (99.999%) powders. The mixed powders were placed in quartz ampoule that was sealed in vacuum with a pressure of 10^−5^ Pa. The quartz ampoule was put into a resistance furnace which was heated at 1170 K for 24 h. Then, it was cooled slowly to and kept at 920 K for 3 days. Finally, it was quenched in cold water. The single crystal was well cleavable. Magnetic measurements were performed using a superconducting quantum-interference device (SQUID, Quantum Design) magnetometer. Field emission scanning electron microscopy (FE-SEM, JSM-7001F) was used to detect the morphology of the samples.

### 2.2. Theoretical Method and Computational Details

The calculations were performed within density functional theory using the Vienna ab initio simulation package (VASP) [57]. The PBE [58] generalized gradient approximation and the projector augmented wave potentials [59] were employed to describe the exchange-correlation functional and the core-electron interaction, respectively. The cut-off energy of all the systems based on the plane wave expansion was set to 500 eV. The integration over the Brillouin zone was done using the Monkhorst–Pack mesh of 3 × 3 × 1. All structures were fully optimized until the convergent threshold for energy was 10^−5^ eV/atom and the Hellmann–Feynman force acting on each atom was 0.01 eV/nm^−1^. The valence electron configuration of each atom was: B 2s^2^2p^1^, C 2s^2^2p^2^, N 2s^2^2p^3^, Se 4s^2^4p^4^, Bi 6s^2^6p^3^. The structure relaxation with DFT-D2 and self-consistent with the spin-orbit coupling (SOC) was calculated. 

## 3. Results and Discussions

### 3.1. Geometrical Structure and Magnetic Properties of X Doping Bi_2_Se_3_

The crystal structure of Bi_2_Se_3_ is rhombohedral, belonging to the space group *D*^5^_3d_ (*R*$\bar{3}$*m*) [60]. As show in Figure 1a, the rhombohedral unit cell contains three Se atoms and two Bi atoms. The hexagonal unit cell has three sets of quintuple layers (QL) with five atomic layers in Figure 1b. The five layers form a stable unit in the order of Se1-Bi-Se2-Bi-Se1 with strong covalent bonds, while the inter-layer bonding is much weaker because of the van der Waals forces. These nonequivalent Se atoms are denoted as Se1 and Se2. The Se2 atom surrounded by Bi atoms is the spatial inversion symmetry center. To study the results of X doping different configurations in Bi_2_Se_3_, we considered the following cases: the Bi site, the non-equivalent Se1 and Se2 sites and the VDW gap site in the 2 × 2 × 1 supercell containing 36 Se atoms and 24 Bi atoms, labeled XBi, XSe1, XSe2 and Xgap. Figure 1c–f are the optimized structure diagrams of Bi_23_Se_36_XBi, Bi_24_Se_35_XSe1, Bi_24_Se_35_XSe2 and Bi_24_Se_36_Xgap, respectively. The change of structure is not particularly obvious with the small doping proportions. The X atom moves close to the Bi and Se1 atoms and forms the X-Se1, X-Bi covalent bond. In Table 1, the strength of covalent bonding increasing accompanies the reducing of the bond length between dopant X and nearby atoms. The bond lengths in the primitive cell are: Se1-Bi, 2.865 Å; Se2-Bi, 3.067 Å and Se1-Se2, 4.36 Å. However, the structure still maintains a high symmetry because the impact on the other QL is inconspicuous.

The formation energy of Bi_2_Se_3_ is given by $E_{{\mathrm{Bi}}_{2}{\mathrm{Se}}_{3}}{=E}_{\mathrm{pure}}\;-\;\left({36\;E}_{\mathrm{Se}}{+\;24\;E}_{\mathrm{Bi}}\right)=-31.362\;\mathrm{eV}$. The formation energy is relevant to the relative stability of structures. Through X doping, the formation energy is computed using the following formulas:(1)$$E_{\mathrm{XBi}}{\;=\;E}_{\mathrm{doped}}{\;-\;E}_{\mathrm{pure}}{\;-\;E}_{X}{\;+\;E}_{\mathrm{Bi}}$$
(2)$$E_{\mathrm{XSe}}{\;=\;E}_{\mathrm{doped}}{\;-\;E}_{\mathrm{pure}}{\;-\;E}_{X}{\;+\;E}_{\mathrm{Se}}$$
(3)$${\;E}_{\mathrm{Xgap}}{\;=\;E}_{\mathrm{doped}}{\;-\;E}_{\mathrm{pure}}{\;-\;E}_{X}$$
where $E_{\mathrm{XBi}}$ is the formation energies of the X atom-doped Bi site; $E_{\mathrm{XSe}}$ is the formation energies of the X atom-doped Se1, Se2 site; $E_{\mathrm{Xgap}}$ represents the formation energies of the X atom-doped gap site; $E_{\mathrm{doped}}$ and $E_{\mathrm{pure}}$ are the total energies of the X-doped and pure Bi_2_Se_3_; and $E_{X}$,${\;E}_{\mathrm{Bi}}$ and $E_{\mathrm{Se}}$ are the energies of the X, Bi and Se atom, respectively. The negative energies mean that X atom-doped Bi_2_Se_3_ can be realized experimentally. As shown in Table 1, the formation energies of the structures with B, C doping at the gap site are lower than those of the structures with substitution at various sites. It is reasonable that those configurations gain large bonding energy due to the strong chemical bonding and then cause a drop in formation energy in these systems. Theoretically, the N atoms with a smaller covalent radius (0.741 Å) (B: 0.905 Å, C: 0.863 Å) are energetically favorable as dopants in Bi_2_Se_3_. The Bi_24_Se_35_NSe1 structure is the most energetically favorable, because the N atom with more free electrons can more easily bond with a Bi atom (bond length: N-Bi: 2.285 Å). In general, the B, C doping gap site and N doping Se1 sites are the most energetically favorable. The X atoms replacing Bi atoms are found to be energetically unfavorable as dopants in Bi_2_Se_3_.

The magnetic properties are calculated because the unpaired electrons are introduced with X atom doping. The ferromagnetic (E_FM_) and antiferromagnetic (E_AFM_) energies with four doped X atoms are calculated to judge the magnetic static state. The *E*_FM_ and *E*_AFM_ in Bi_24_Se_32_X_4_ are −248.801 eV and −248.786 eV, −276.604 eV and −273.49 eV, −254.697 eV and −254.521 eV, respectively. This demonstrates that the ferromagnetic interaction is more favorable than the antiferromagnetic one, and the system of X doping Bi_2_Se_3_ is ferromagnetic. Xin [56] reported the single crystal Bi_2_C_x_Se_3−x_ (x = 0.05) sample is a FM state which excludes magnetic impurities. This means that non-magnetic atoms can introduce magnetism in Bi_2_Se_3_.

The total magnetic moments (M_tot_) and single X atom moments (M_X_) are listed in Table 1. The ferromagnetism mainly comes from the doped atoms, and the doped atoms also have an effect on the surrounding atoms. The magnetic moments of BBi, BSe2, CBi, CSe1, CSe2, Cgap, NSe2 and Ngap are 0.978 μB, 1.416 μB, 0.410 μB, 0.649 μB, 0.980 μB, 0.673 μB, 0.094 μB and 0.794 μB, respectively. As illustrated in our results, the magnetic moments are generally reasonable and closer to the experimental data. Specifically, the bond lengths decrease and the covalent bonds become stronger with X atoms doping. There are more charges transferring to the dopant X from adjacent atoms and occupying the empty 2p orbitals of the dopant. The 2p orbital of the dopant atoms become less localized or even delocalized. The bond lengths and transferred charges are different when X dopes different sites. The magnetic moments’ variation accompanies the change of charges. Thus, the magnetic moments and the bond lengths vary for different sites in this case. It is noted the magnetic moment can be introduced when C atoms are doped at each position, suggesting that C is the most effective dopant. In addition, the X atom doping at the Se2 site always introduces the magnetic moment, and the magnetic moment is larger than that of the Se1 site. In the experiment, it is difficult to control the doping element to replace the Se2 site definitely, so the best way to introduce magnetism is through C atom doping.

### 3.2. Electronic Structure of C Doping Bi_2_Se_3_

Next, we focus on the electronic structure of C doping Bi_2_Se_3_ because C doping can introduce magnetism at each doping position. The valence band maximum (VBM) of the perfect Bi_2_Se_3_ is dominated by the p states of Se1 and Se2, whereas the conduction band minimum originates (CBM) from Bi-6p without SOC. Due to the effect of spin orbit coupling, the VBM mainly derives from the p states of the 6p electrons of the Bi atom; the 4p electrons of the Se atom shift from VBM to CBM. The energy band reversed at the Γ point with SOC, as shown in Figure 2. Figure 3 is the band structures of C atoms doping Bi_2_Se_3_ with the SOC functional. The Fermi level is set at the energy zero point and the band structure (−1~1 eV) close to the Fermi level is given. 

The impurity energy band mainly originating from the doped atom is localized near the Fermi level. This shows that the energy band of Bi_2_Se_3_ can be regulated by non-magnetic atoms doping. The impurity states mixing with the time-reversal paired states resulted in the substantial modifications of the electronic structure. Even when the contribution of dopants in the bands is very weak, the dopants have a very large impact on the band gap, band curvature and Fermi level, varying wildly depending on the position of the dopant. To make the impurity states more obvious, the contribution of the C atoms was magnified 10 times.

The C substitution for Se created a hole carrier because Bi atoms can accommodate three electrons but there are not enough electrons for the C atom to bond with the surrounding Bi atoms. Thus, in Figure 3b,c, the impurity band near the Fermi level mainly derives from the hole carrier. Moreover, because Se2 atoms are in the center of the QL and strongly localized, there is more than one impurity band between the valence band and the conduction band. In Figure 3a,d, the Fermi level moves up into the conduction band. The Fermi level undergoes an evident shift in energy, presenting somewhat metallic behavior, and the insulation properties are damaged. In Cgap, the impurity bands appear in the gap to accommodate the introduced unpaired electrons. The energy degeneracy is more obvious because the van der Waals force is weaker than the covalent bond. Similar to pure Bi_2_Se_3_, the energy band reversed at the Γ point, indicating that the doped system still maintains topological properties. The conduction band is contributed by Bi-p states. The valence band is derived from the Se-p states and the C-2p orbital. The bottom of the conduction band at the Γ point still mainly comes from the p-electron contribution of the Se atom, and the valence band top comes from the p-electron contribution of the Bi atom.

Figure 4 shows the total density of states (DOS) and partial density of states (PDOS) after doping. To make the impurity states more obvious, the intensity of the integral C-2p states was 30 times larger. The narrow highly localized impurity bands exhibiting an obvious spin split around the Fermi level. The spin-up and spin-down densities of states are offset and asymmetrical, indicating the appearance of magnetism. In CSe1 and CSe2, it is obvious that for the spin-up channel there is an energy gap around the Fermi level while the minority spin-down bands cross the Fermi level. The substituted structures are half-metallic ferromagnetic. This is different from the CBi and Cgap, where the majority spin channel is metallic.

In order to verify the magnetism of C-doped samples, single crystal samples were prepared and magnetization versus applied magnetic field curves for C_x_Bi_2_Se_3_ was measured at 15 K. The estimated diamagnetic susceptibility was about *χ*_0_ = −8.22 × 10^−5^ emu/mol, which is close to that of pure Bi_2_Se_3_. Similar diamagnetic signals are also obvious in the C-doped samples, as shown in Figure 5a. After subtraction of the paramagnetic signals from the total signals, hysteresis loops were extracted from the experimental data [35]. As can be seen in Figure 5b, the magnetic saturation moment per C atom was 0.0235~0.0375 μB, as listed in Table 2. In spite of supposing a very low amount of C atom in Bi_2_Se_3_, it is also possible to measure the coercivity *H_c_* (shown in Table 2). As the concentration of C increases, the *M*_Smol_ values increase obviously. The coercive field *H_c_* varies from 112 Oe to 232 Oe, nearly invariant of the C concentration. It is noted that the remnant magnetization *M*_r_ increases with the C concentration. It can be supposed that ferromagnetism is closely related to C concentration, based on the rising tendency of *M*_Smol_ and *M*_r_. The experimental magnetic moment of C_x_Bi_2_Se_3_ is considerably smaller than the expected atomic moments (shown in Table 1). This may be due to the uneven doping during the preparation process, as only a small part of C enters the lattice. There are clear hysteresis loops at 15 K, which indicates the existence of a ferromagnetic state in C_x_Bi_2_Se_3_.

## 4. Conclusions

In the present work, the electrical and magnetic properties of the C-doped Bi_2_Se_3_ topologic insulator were discussed. The Fermi level moves up into the conduction band in CBi and Cgap and the insulation properties of the Bi_2_Se_3_ system are damaged. In CSe1 and CSe2, the substituted structures are half-metallic ferromagnetic. Similar to the transition metals doping, the X atom doping Bi_2_Se_3_ can induce magnetic moments. It was proved that C is the most effective dopant to induce the magnetism ground states. The remnant magnetization Mr increases with the C concentration. The experiment confirmed the existence of ferromagnetic state in C_x_Bi_2_Se_3_. The introduction of magnetism by doping with non-magnetic elements can promote the application of topological insulators in spintronic devices.

## Figures and Tables

**Figure 1 materials-15-03864-f001:**
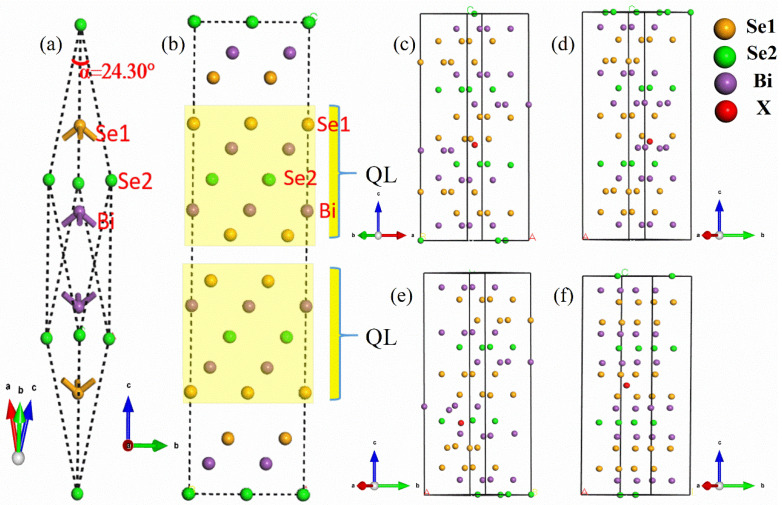
(**a**) The rhombohedral unit cell of Bi_2_Se_3_, (**b**) 60-atom layered crystal structure of Bi_2_Se_3_, (**c**) structure of Bi_23_Se_36_XBi, (**d**) structure of Bi_24_Se_35_XSe1, (**e**) structure of Bi_24_Se_35_XSe2, and (**f**) structure of Bi_24_Se_36_Xgap.

**Figure 2 materials-15-03864-f002:**
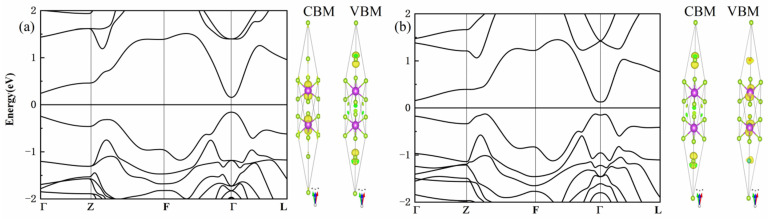
Band structure and partial charge of Bi_2_Se_3_: (**a**) GGA and (**b**) SOC.

**Figure 3 materials-15-03864-f003:**
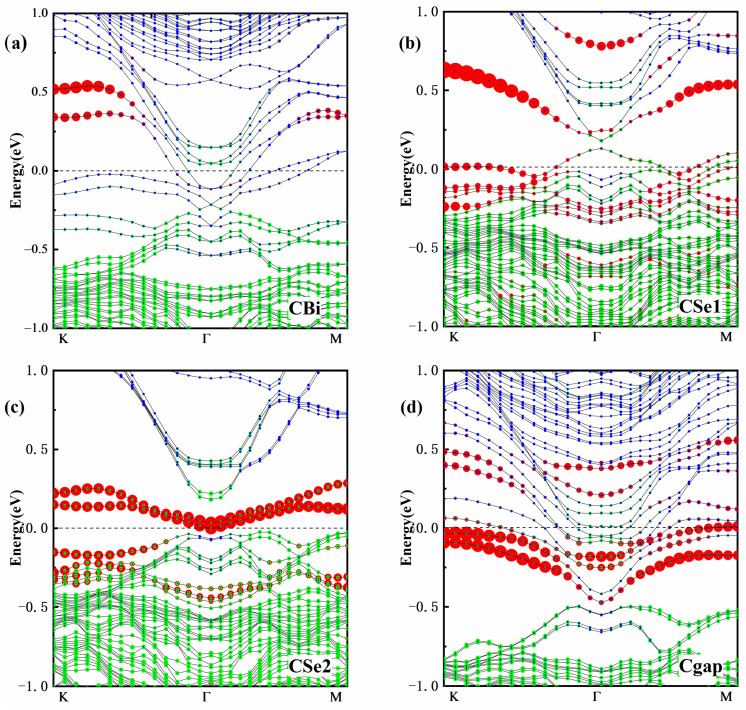
Band structure of C doping Bi_2_Se_3_: (**a**) C doping at Bi site, (**b**) C doping at Se1 site, (**c**) C doping at Se2 site, and (**d**) C doping at gap site. The black line is the total band, the red point is the contribution from C atom, the green point is the contribution from Se atom and the blue point is the contribution from Bi atom.

**Figure 4 materials-15-03864-f004:**
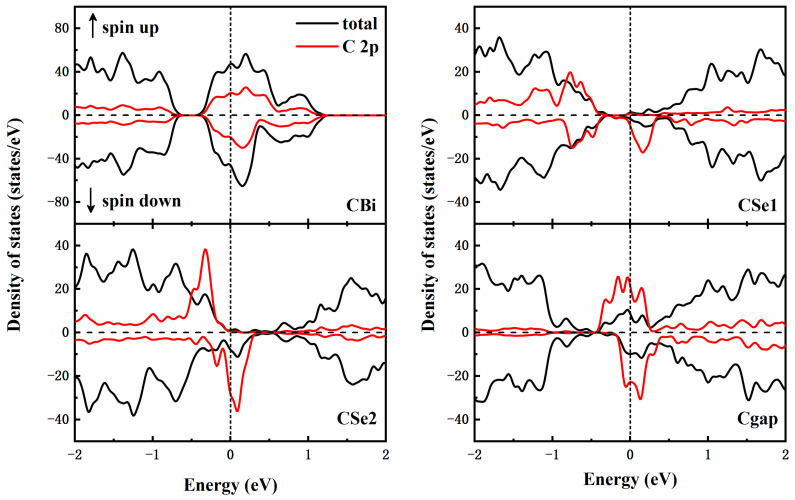
The total DOS and PDOS of C doping Bi_2_Se_3_.

**Figure 5 materials-15-03864-f005:**
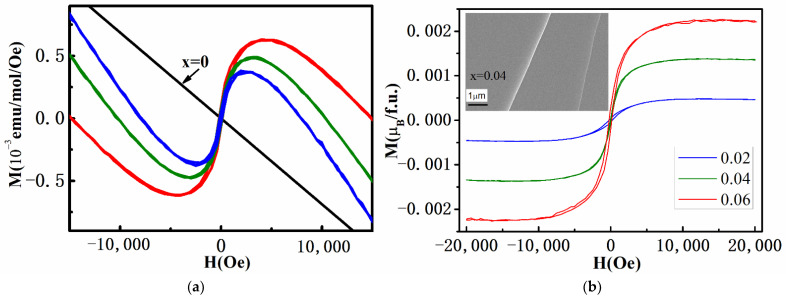
(**a**) Magnetization versus applied magnetic field for C_x_Bi_2_Se_3_ measured at 15 K. (**b**) The magnetic hysteresis loops after subtracting the diamagnetic signals. There is obvious magnetic hysteresis in all doped samples. Inset: FESEM images of C_0.06_Bi_2_Se_3_ crystal.

**Table 1 materials-15-03864-t001:** Formation energies, bond lengths and magnetic moments of X doping Bi_2_Se_3_.

	*E*_from_ (eV)	*d*_X-Bi_ (Å)	*d*_X-Se_ (Å)	M_tot_ (μB)	M_X_ (μB)	M_tot_ (μB)._exp_
BBi	−4.565	/	1.979	0.978	0.208	
BSe1	−11.007	2.460	/	0	0	0.44 ^a^
BSe2	−10.52	2.838	/	1.416	0.413	3.0 ^a^
Bgap	−12.177	/	1.832	0	0	
CBi	−0.808	/	1.885	0.410	0.164	
CSe1	−9.933	2.369	/	0.649	0.377	2.0 ^a^, 1.66 ^b^
CSe2	−9.492	2.798	/	0.980	0.375	2.0 ^a^, 1.95 ^b^
Cgap	−9.975	/	1.801	0.673	0.225	1.53 ^b^
NBi	−16.216	/	1.997	0	0	
NSe1	−17.418	2.285	/	0	0	0.14 ^a^
NSe2	−17.151	2.325	/	0.094	0.028	1.0 ^a^
Ngap	−16.303	/	1.956	0.794	0.576	

^a^ Data taken from Ref. [13]. ^b^ Data taken from Ref. [56].

**Table 2 materials-15-03864-t002:** Lists of remnant magnetization, saturation magnetization and coercivity for C_x_Bi_2_Se_3_.

x	Saturation Magnetization M_Smol_		Remanent Magnetization M_r_ [μB/f.u.]	Coercivity *H**_c_* (Oe)
	[μB/f.u.]	[μB/C_atom_]		
0.02	4.71 × 10^−4^	0.0235	7.85 × 10^−5^	227
0.04	1.37 × 10^−3^	0.034	2.1 × 10^−4^	112
0.06	2.25 × 10^−3^	0.0375	7.5 × 10^−4^	232

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
