# Peer review of "An Efficient Dopant for Introducing Magnetism into Topological Insulator Bi2Se3"

_materials, 2022, doi:10.3390/ma15113864_

Round 1
Reviewer 1 Report
The structural, electrical, and magnetic properties of Bi2Se3 doped with carbon at Bi, Se1, Se2, and VDW bandgap sites were studied by the pseudopotential plane wave method of density functional theory. It was demonstrated that carbon doping for Bi2Se3 can cause appearance of impurity bands in the bandgap. It was shown that the remnant magnetization increases with increasing carbon concentration. The Authors claim that their experiment confirms the existence of ferromagnetic state in CxBi2Se3. However, there are several weak points of the current studies that must be clarified prior any decision concerning publication of the manuscript:
(1) Possibility of the appearance of carbon dimers or trimers in carbon doped Bi2Se3 is completely ignored in the manuscript despite that it was already shown that the formation energies of the structures with C-dimer substitution for Se are much lower (over 2.5 eV) than those of the structures with isolated carbon substitution at various sites (see, for example, Xiaojun Xin, Chunsheng Guo, Rui Pang, Min Zhang, Xingqiang Shi, Xinsheng Yang, and Yong Zhao, Appl. Phys. Lett. 115, 042401 (2019); https://doi.org/10.1063/1.5110182).
(2) Figure 4 presents magnetization versus applied magnetic field for CxBi2Se3 with x = 0.02, 0.04, and 0.004. However, there is a luck of any information in the manuscript how the studied samples were prepared. It will be useful to add images characterizing quality of the samples – for example field emission scanning electron microscopy images.
(3) There is luck of any information how the background signal was subtracted to get magnetization data presented in Figure 4. Such information is very important if the Authors want to convince readers that their data are reliable.
(4) There is lack of any evidence supporting the statement “There are clear hysteresis loops at 2.1 K which indicates the existence of ferromagnetic state in CxBi2Se3” given in the manuscript. Could the Authors present experimental data? How hysteresis loop changes with increasing temperature?
(5) The English of the current version of the manuscript is unacceptable. Several sentences are non-understandable.
Author Response
(1) Possibility of the appearance of carbon dimers or trimers in carbon doped Bi2Se3 is completely ignored in the manuscript despite that it was already shown that the formation energies of the structures with C-dimer substitution for Se are much lower (over 2.5 eV) than those of the structures with isolated carbon substitution at various sites (see, for example, Xiaojun Xin, Chunsheng Guo, Rui Pang, Min Zhang, Xingqiang Shi, Xinsheng Yang, and Yong Zhao, Appl. Phys. Lett. 115, 042401 (2019); https://doi.org/10.1063/1.5110182).
Answer: Thanks a lot for your kind suggestion. We agree the formation energies of the structures with C-dimer substitution for Se are much lower (over 2.5 eV) than those of the structures with isolated carbon substitution at various sites. But dimer- and trimer- substitutions for Se which induce neither magnetism nor charge doping. In fact, the isolated, pairwise, and trimeric carbon dopants should coexist in the sample. The fabricated single crystal C-Bi2Se3 is ferromagnetic, indicated there is isolated C element. Here, we focus on the discussions of isolated carbon doping at Bi2Se3 to explore the introducing magnetism into topological insulator.
(2) Figure 4 presents magnetization versus applied magnetic field for CxBi2Se3 with x = 0.02, 0.04, and 0.004. However, there is a lack of any information in the manuscript how the studied samples were prepared. It will be useful to add images characterizing quality of the samples – for example field emission scanning electron microscopy images.
Answer: Detailed sample preparation process in the manuscript. Please see "2. Experimental and computational details”. Field emission scanning electron microscopy image has been added in Figure 4.
(3) There is lack of any information how the background signal was subtracted to get magnetization data presented in Figure 4. Such information is very important if the Authors want to convince readers that their data are reliable.
Answer: The estimated diamagnetic susceptibility is about χ0=-8.22×10-5 emu/mol, which is close to that of pure Bi2Se3. Similar diamagnetic signals are also obvious in C-doped samples, shown in Figure. 4(a). Li et al [1] explained that the diamagnetic signal originates from phase separation of non-magnetic Bi2Se3 and magnetic phase. After subtraction of the paramagnetic signals from the total signals, hysteresis loops have been extracted from the experimental data.
[1] H. Li et al., AIP J. J. Appl. Phys. 113, 043926 (2013).
(4) There is lack of any evidence supporting the statement “There are clear hysteresis loops at 2.1 K which indicates the existence of ferromagnetic state in CxBi2Se3” given in the manuscript. Could the Authors present experimental data? How hysteresis loop changes with increasing temperature?
Answer: Thanks for reminding us. The “2.1” have already been modified to “15K” in the revised manuscript. In order to verify the magnetism of C-doped samples, single crystal samples were prepared and magnetization versus applied magnetic field curves for CxBi2Se3 has been measured at 15 K. The changes of hysteresis loop with increasing temperature with be measure in follow work.
(5) The English of the current version of the manuscript is unacceptable. Several sentences are non-understandable.
Answer: Thanks a lot for your careful comments. Some writing issues have been solved and this manuscript undergoes extensive English revisions.

Reviewer 2 Report
The manuscript explains the structural, electrical, and magnetic properties of non-magnetic element C doped at Bi, 9 Se1, Se2 and VDW bandgap sites of Bi2Se3 using the first-principles pseudopotential plane-wave method of density functional theory (DFT). Also, presented the experimental results and discussed them. The study has potential and is worth reporting. However, the manuscript should undergo significant amendments before being accepted for journal publication. Please see the specific comments as follows.
Abstract
The abstract should be rewritten by providing and highlighting major results
Introduction
The intro should rearrange to highlight the motivation and aims of the study
Method
Should provide the details of making the compound by mixing powders. Which ratio of powders etc....? It will help to reproduce the compound
Comments for simulation work
Figure 2, Should include the Bi2Se3 band structure for the comparison
Comments for experimental work
- How did the author claim the compound as a single crystal material? Should provide enough characterisations to justify the claim (XRD, TEM etc...)
- Also, should provide either XRF or EDX to confirm the presence of all elements in the compound
Author Response
The manuscript explains the structural, electrical, and magnetic properties of non-magnetic element C doped at Bi, 9 Se1, Se2 and VDW bandgap sites of Bi2Se3 using the first-principles pseudopotential plane-wave method of density functional theory (DFT). Also, presented the experimental results and discussed them. The study has potential and is worth reporting. However, the manuscript should undergo significant amendments before being accepted for journal publication. Please see the specific comments as follows.
(1)Abstract
The abstract should be rewritten by providing and highlighting major results
Answer:
Thanks a lot for your careful suggestion. The abstract has be corrected with providing and highlighting major results.
(2)Introduction
The intro should rearrange to highlight the motivation and aims of the study.
Answer:
We rearrange the introduction to highlight the motivation and aims of the study.
(3)Method
Should provide the details of making the compound by mixing powders. Which ratio of powders etc....? It will help to reproduce the compound
Answer:
The single crystalline CxBi2Se3 (x=0, 0.02, 0.04, 0.06) were prepared from the reactions of the stoichiometric mixture of carbon (99.99%), Bi (99.999%) and Se (99.999%) powders. The mixed powders were placed in quartz ampoule that is sealed in vacuum with a pressure of 10-5Pa. The quartz ampoule was put into a resistance furnace which is heated at 1170 K for 24 hours. Then it was cooled slowly to and kept 920 K for 3 days. Finally quenched in cold water.
(4)Comments for simulation work
Figure 2, Should include the Bi2Se3 band structure for the comparison
Answer:
The band structure of Bi2Se3 is shown in Figure 2.
(5)Comments for experimental work
- How did the author claim the compound as a single crystal material? Should provide enough characterisations to justify the claim (XRD, TEM etc...)
- Also, should provide either XRF or EDX to confirm the presence of all elements in the compound
Answer:
Fig. 1. (a)X-ray diffraction patterns of CxBi2Se3 with different C concentration (x=0 and 0.06). (b)The full width at half maximum of the (0 0 6) peak is less than 0.02°. FESEM images for CxBi2Se3-x with different C concentration: (c) x=0 and (d) x=0.06.
The θ-2θ x-ray diffraction pattern of CxBi2Se3 (x=0, 0.06) crystals are shown in Fig. 1(a). The (0 0 L) family of diffraction peaks are observed with no obvious impurity phase, indicating that the crystals are exclusively single crystallinity. Rocking wave datum of C0.06Bi2Se3show that the full width at half maximum (FWHM) of the (0 0 6) peak is 0.069° shown in Fig.1(b), which indicating that the samples are highly c-axis oriented along the growth direction with high crystal quality. The lattice constant c varies from 28.64Å for non-doped sample to 28.61Å for x=0.06. Fig. 1(c) and (d) show FESEM images of natural cleavage planes of undoped Bi2Se3 and doped Bi2Se3 crystals. The layered structure and steps can be clearly observed, validating the trigonal-axis oriented with excellent crystal quality. Elemental compositional analysis of a series of the CxBi2Se3 samples is listed in Table 1, showing that the percentage of Bi/Se atoms is close to the ideal ratio with a small amount of Se vacancies. The quantitative analysis of EDX results again verifies the excellent crystal quality.
Table 1 The stoichiometric and experimental composition of CxBi2Se3 samples with x=0, and 0.06 measured by EDX analysis
|
Sample |
Stoichiometric(at.%) |
Experimental(at.%) |
||||
|
C |
Bi |
Se |
C |
Bi |
Se |
|
|
0.0 |
0 |
40 |
60 |
0 |
40.02 |
59.98 |
|
0.06 |
1.6 |
40 |
58.4 |
1.73 |
40.98 |
57.29 |

Reviewer 3 Report
In this study, light metal dopants (C, N and B) are introduced to Bi2Se3, which is a 3D topological insulator. It had been demonstrated previously that Bi2Se3 doped by C, N and B were ferromagnetic [Applied Physics Letters 100.25 (2012): 252410.]. However, the dopant only replaced Se sites. In another work other sites were studied but only for C impurities [Applied Physics Letters 115.4 (2019): 042401.]. In this work 4 different sites were considered for C,N and B impurities in Bi2Se3. Using first principles calculations, the authors studied whether these impurities will induce magnetic moment.
The structural properties are reported. The electronic band structure and DOS are plotted with the dopants. The authors calculated single atom magnetic moment as well as total magnetic moment and mentioned that C dopant produces the most magnetism. They also reported their experimental results which demonstrated magnetism as a function of magnetic field and concluded that magnetism and remnant magnetism increase with doping concentration.
The writing manuscript must be improved in regard to English.
- To improve the novelty and interest of this manuscript, more information should be provided about the topological phase transition.
For instance, it was mentioned that the band inversion still occurs when the dopants are added. It raises the question that whether the occurrence of band inversion is sufficient to ensure the topologically nontrivial phase. Additionally, the band inversion is not easily detectable in figure 2 for all the doping sites. Also, what is expected to happen if the doping concentration goes higher in terms of topological phase transition?
- What is the effect of doping on the surface states? in figure 2, it’s not clear if the band structure is plotted for the bulk Bi2Se3 or terminated Bi2Se3. in the method section of the paper, it is specified that a 3*3*1 k-mesh sampling is utilized. If it indicates that the band structure is plotted for a terminated TI with surface states, it would be beneficial to plot the band structure for the case without dopant. So, the opening of the gap in the Dirac cones as a result of the doping would be apparent. However, it appears that the bulk band structure is plotted in figure 2. So, is the use of Monkhorst-Pack mesh of 3×3×1 sufficient? Also, it can be beneficial to discuss the effects of the added impurity on the surface states and bulk gaps.
- There are a few papers that have investigated the effects of light metal doping (such as C,N and B) in Bi2Se3. The comparisons of the results could be added to the manuscript. For instance, in paper “Xin, Xiaojun, et al. "Theoretical and experimental studies of spin polarized carbon doped Bi2Se3." Applied Physics Letters 115.4 (2019): 042401.” total magnetic moments for Bi2Se3 with C doping at various sites are reported. For other dopants (N and B) the results could be compared with “Niu, Chengwang, et al. "Ferromagnetism and manipulation of topological surface states in Bi2Se3 family by 2p light elements." Applied Physics Letters25 (2012): 252410” . It is stated in the manuscript that “Xin [56] reported the single crystal Bi2CxSe3-x (x=0.05) sample is FM state which excludes magnetic impurities. It means that nonmagnetic atom can introduce magnetism in Bi2Se3. “. However, a more detailed comparison in terms of the calculated values is missing.
- It is mentioned in the paper that “The magnetic properties are different with different atoms doping at different positions.” That is also demonstrated in table. 1. Could you explain why the magnetic moments (or the bond lengths) vary for different sites in this case?
- It is mentioned in the paper that the remnant magnetization and the saturation
magnetization increase with C concentration. Do you expect an upper limit? Do you expect a magnetic phase transition if the doping concentration goes higher?
- Is the magnetism affected by the topological phase and the band inversion with these group of dopants? Is the magnetism still present if SOC is turned off and band inversion is ignored?
- Several papers have explored the effects of transition metals as dopants in Bi2Se3. Based on your results, what are the benefits or disadvantages of using light metal dopants instead of transition metals?
- Some minor suggestions:
Maybe a figure can show an image of the crystal that is used for the experimental part of the paper.
Bismuth is a heavy element. Is the chosen valence electron configuration for DFT simulations sufficient?
The xyz directions in figure 1 are not clarified
Author Response
(1) The writing manuscript must be improved in regard to English.
Answer: Thanks a lot for your kind suggestion. We have revised the whole manuscript carefully to avoid grammar errors and sought assistance from colleague to correct the language in our manuscript. We hope that the language is now acceptable for further review process.
(2) To improve the novelty and interest of this manuscript, more information should be provided about the topological phase transition.
For instance, it was mentioned that the band inversion still occurs when the dopants are added. It raises the question that whether the occurrence of band inversion is sufficient to ensure the topologically nontrivial phase. Additionally, the band inversion is not easily detectable in figure 2 for all the doping sites. Also, what is expected to happen if the doping concentration goes higher in terms of topological phase transition?
Answer: Thanks a lot for your kind suggestion. According to your comments, we have highlighted the topological phase transition of this work in our revised manuscript. We think the occurrence of band inversion is sufficient to ensure the topologically nontrivial phase, more detailed evidence will be provided in the follow work. In CSe1 and CSe2, it is obvious that for the spin up channel there is an energy gap around the Fermi level while the minority spin-down bands cross the Fermi level. The substituted structures are half-metallic ferromagnetic. This is different with the CBi and Cgap, where the majority spin channel is metallic. The half-metallic or metallic properties will enhance if the doping concentration goes higher with more localized electron.
(3) What is the effect of doping on the surface states? in figure 2, it’s not clear if the band structure is plotted for the bulk Bi2Se3 or terminated Bi2Se3. in the method section of the paper, it is specified that a 3*3*1 k-mesh sampling is utilized. If it indicates that the band structure is plotted for a terminated TI with surface states, it would be beneficial to plot the band structure for the case without dopant. So, the opening of the gap in the Dirac cones as a result of the doping would be apparent. However, it appears that the bulk band structure is plotted in figure 2. So, is the use of Monkhorst-Pack mesh of 3×3×1 sufficient? Also, it can be beneficial to discuss the effects of the added impurity on the surface states and bulk gaps.
Answer: Considering your suggestion, we have concerned the effects of the added impurity on the surface states. It is stated in the paper “Shen, L., et al. "Simultaneous Magnetic and Charge Doping of Topological Insulators with Carbon." Physical Review Letters 111.23(2013):236803”, the effect of C doping on the surface states was discussed. It is known that, the surface state of Bi2Se3 is metallic. Shen have demonstrated simultaneous magnetic and hole doping achieved with a single dopant, carbon, in Bi2Se3 by first-principles calculations. Carbon substitution for Se (CSe) results in an opening of a sizable surface Dirac gap (up to 82 meV), while the Fermi level remains inside the bulk gap and close to the Dirac point at moderate doping concentrations. The strong localization of 2p states of CSe favors spontaneous spin polarization via a p-p interaction and formation of ordered magnetic moments mediated by surface states. Meanwhile, we have calculated the (001) surface with C doping on gap site, as shown in Figure S1. The results show the surface state still is metallic, because free electrons are introduced into the system by Cgap.
Figure S1. Band structure of C doping (001) surface of Bi2Se3.
In figure 2, the band structures are plotted for the bulk Bi2Se3 with C doping. The lattice constants a=b=8.29Å, c=28.6 Å. Because the length of c-axis is larger than the a-axis and b-axis, the Monkhorst-Pack mesh of 3×3×1 is not for a terminated TI with surface states. The k-mesh sampling is produced by VASPKIT with k-mesh-resolved value of 0.04 and generally precise enough to guarantee to converge.
(4) There are a few papers that have investigated the effects of light metal doping (such as C, N and B) in Bi2Se3. The comparisons of the results could be added to the manuscript. For instance, in paper “Xin, Xiaojun, et al. "Theoretical and experimental studies of spin polarized carbon doped Bi2Se3." Applied Physics Letters 115.4 (2019): 042401.” total magnetic moments for Bi2Se3 with C doping at various sites are reported. For other dopants (N and B) the results could be compared with “Niu, Chengwang, et al. "Ferromagnetism and manipulation of topological surface states in Bi2Se3 family by 2p light elements." Applied Physics Letters25 (2012): 252410”. It is stated in the manuscript that “Xin [56] reported the single crystal Bi2CxSe3-x (x=0.05) sample is FM state which excludes magnetic impurities. It means that nonmagnetic atom can introduce magnetism in Bi2Se3. “. However, a more detailed comparison in terms of the calculated values is missing.
Answer: Thanks a lot for your kind suggestion. The comparisons in terms of the magnetism are added to the manuscript and more detailed results of the magnetic moments are shown in Table. 1.
(5) It is mentioned in the paper that “The magnetic properties are different with different atoms doping at different positions.” That is also demonstrated in table. 1. Could you explain why the magnetic moments (or the bond lengths) vary for different sites in this case?
Answer: Specifically, the bond lengths decrease and the covalent bonds become stronger with X atoms doping. There are more charges transfer to the dopant X from adjacent atoms and occupy the empty 2p orbitals of the dopant. The 2p orbital of the dopant atoms become less localized or even delocalized. The bond lengths and transferred charges are different when X dope different sites. The magnetic moments vary accompanies the change of charges. So the magnetic moments (or the bond lengths) vary for different sites in this case.
(6) It is mentioned in the paper that the remnant magnetization and the saturation
magnetization increase with C concentration. Do you expect an upper limit? Do you expect a magnetic phase transition if the doping concentration goes higher?
Answer: Theoretical calculation indicates that the largest magnetic moments are equal to the numbers of holes in systems [1]. The structure with be destroyed if the doping concentration goes higher.
[1] Niu C W. et al., Appl. Phys. Lett.100.25:194 (2012).
(7) Is the magnetism affected by the topological phase and the band inversion with these group of dopants? Is the magnetism still present if SOC is turned off and band inversion is ignored?
Answer: Many thanks for your kind suggestion. According to your comment, we have calculated the magnetism without SOC. The results show the magnetism still present when SOC is turned off and band inversion is ignored. But magnetic moments are reduced, for example, the magnetic moment of Cgap changes from a value of 0.673μB to 0.457μB.
(8) Several papers have explored the effects of transition metals as dopants in Bi2Se3. Based on your results, what are the benefits or disadvantages of using light metal dopants instead of transition metals?
Answer: The nonmagnetic materials become ferromagnetic due to light elements doping. An obvious advantage is that the clusters or secondary phases formed by the dopants do not contribute to magnetism comparing with conventional magnetic semiconductors. In particular, it is shown that carbon substitutions for Se in Bi2Se3 simultaneously introduced localized magnetic moments and holes. As a result, the CSe dopants lead to the changes of the Dirac gap and pinning of the Fermi level inside the bulk energy gap. The carbon dopants in gap introduces free electrons, resulting in shifting up in energy of the Fermi level. Based on our results, which is benefited for striking physics and device applications for topological insulators.
(9) Some minor suggestions: Maybe a figure can show an image of the crystal that is used for the experimental part of the paper.
Bismuth is a heavy element. Is the chosen valence electron configuration for DFT simulations sufficient?
The xyz directions in figure 1 are not clarified
Answer: Special thanks for your good comments and suggestions. The FESEM image of C0.06Bi2Se3 crystal is shown in the inset of Figure 4. (b).
The impurity bands appear in the gap to accommodate the change of the charges and induces magnetism. The changed charges are extra-nuclear electron, so the chosen valence electron (6s26p3) for Bi element is sufficient to simulate the electronic structure.
The xyz directions in figure 1 are clarified.

Round 2
Reviewer 1 Report
The manuscript has been improved. Despite that I am still not fully satisfied with its content, I can accept the manuscript in its current form and therefore, I recommend it for publication in Materials.
Author Response
Thank you for your approval. We will continue to improve the papers before publication.Reviewer 2 Report
Thank you for revising the manuscript. It has improved after the amendments. However, my 2nd and 5th comment did not answer yet.
2nd commet:- Please check the intro and revise as suggested before.
5th commet:- The authors have replied with the following, but I don’t see the mentioned Fig 1 showing XRD pattern and SEM images in the revised manuscript. Also, table 1 in the manuscript is not the same as here. Please check and revise again accordingly to answer the 5th comment.
5)Comments for experimental work
- How did the author claim the compound as a single crystal material? Should provide enough characterisations to justify the claim (XRD, TEM etc...)
- Also, should provide either XRF or EDX to confirm the presence of all elements in the compound
Answer:
Fig. 1. (a)X-ray diffraction patterns of CxBi2Se3 with different C concentration (x=0 and 0.06). (b)The full width at half maximum of the (0 0 6) peak is less than 0.02°. FESEM images for CxBi2Se3-x with different C concentration: (c) x=0 and (d) x=0.06.
The θ-2θ x-ray diffraction pattern of CxBi2Se3 (x=0, 0.06) crystals are shown in Fig. 1(a). The (0 0 L) family of diffraction peaks are observed with no obvious impurity phase, indicating that the crystals are exclusively single crystallinity. Rocking wave datum of C0.06Bi2Se3show that the full width at half maximum (FWHM) of the (0 0 6) peak is 0.069° shown in Fig.1(b), which indicating that the samples are highly c-axis oriented along the growth direction with high crystal quality. The lattice constant c varies from 28.64Å for non-doped sample to 28.61Å for x=0.06. Fig. 1(c) and (d) show FESEM images of natural cleavage planes of undoped Bi2Se3 and doped Bi2Se3 crystals. The layered structure and steps can be clearly observed, validating the trigonal-axis oriented with excellent crystal quality. Elemental compositional analysis of a series of the CxBi2Se3 samples is listed in Table 1, showing that the percentage of Bi/Se atoms is close to the ideal ratio with a small amount of Se vacancies. The quantitative analysis of EDX results again verifies the excellent crystal quality.
Table 1 The stoichiometric and experimental composition of CxBi2Se3 samples with x=0, and 0.06 measured by EDX analysis
|
Sample |
Stoichiometric(at.%) |
Experimental(at.%) |
||||
|
C |
Bi |
Se |
C |
Bi |
Se |
|
|
0.0 |
0 |
40 |
60 |
0 |
40.02 |
59.98 |
|
0.06 |
1.6 |
40 |
58.4 |
1.73 |
40.98 |
57.29 |
Author Response
Reviewer #2:
2nd commet:- Please check the intro and revise as suggested before.
Answer:
The introduction have been rewritten.
5th commet:- The authors have replied with the following, but I don’t see the mentioned Fig 1 showing XRD pattern and SEM images in the revised manuscript. Also, table 1 in the manuscript is not the same as here. Please check and revise again accordingly to answer the 5th comment.
- How did the author claim the compound as a single crystal material? Should provide enough characterisations to justify the claim (XRD, TEM etc...)
- Also, should provide either XRF or EDX to confirm the presence of all elements in the compound
Answer:
Reviewer #2:
2nd commet:- Please check the intro and revise as suggested before.
Answer:
The introduction have been rewritten.
5th commet:- The authors have replied with the following, but I don’t see the mentioned Fig 1 showing XRD pattern and SEM images in the revised manuscript. Also, table 1 in the manuscript is not the same as here. Please check and revise again accordingly to answer the 5th comment.
- How did the author claim the compound as a single crystal material? Should provide enough characterisations to justify the claim (XRD, TEM etc...)
- Also, should provide either XRF or EDX to confirm the presence of all elements in the compound
Answer:
The XRD pattern, SEM images and EDX analysis are shown in next Fig. S1and Table S1. We have provided these data in this reply, but we are sorry that we cannot provide them in the manuscript. These data have been reported in other papers we recently submitted.
The XRD pattern, SEM images and EDX analysis are shown in next Fig. S1and Table S1. We have provided these data in this reply, but we are sorry that we cannot provide them in the manuscript. These data have been reported in other papers we recently submitted.

Reviewer 3 Report
Thank you for your answer. I do not have any further objections.
Author Response
Thank you for your approval. We will continue to improve the paper before publication.